# Location Dictates Snow Aerodynamic Roughness

**Steven R. Fassnacht** [1,2,*], **Kazuyoshi Suzuki** [3], **Masaki Nemoto** [4], **Jessica E. Sanow** [1], **Kenji Kosugi** [4], **Molly E. Tedesche** [5,6] and **Markus M. Frey** [7]

1  ESS-Watershed Science, Colorado State University, Fort Collins, CO 80523-1476, USA; jessica.sanow@colostate.edu
2  Cooperative Institute for Research in the Atmosphere, Fort Collins, CO 80523-1375, USA
3  Japan Agency for Marine-Earth Science and Technology (JAMSTEC), 3173-25 Showamachi, Kanazawa-ku, Yokohama 236-0001, Kanagawa, Japan; skazu@jamstec.go.jp
4  Shinjo Cryospheric Environment Laboratory, Snow and Ice Research Center, National Research Institute for Earth Science and Disaster Resilience, Shinjo 996-0091, Yamagata, Japan; mnemoto@bosai.go.jp (M.N.); kosugi@bosai.go.jp (K.K.)
5  Cold Regions Research & Engineering Laboratory, US Army Corps Engineer Research & Development Center, 72 Lyme Rd., Hanover, NH 03755-1290, USA; metedesche@alaska.edu
6  Institute of Northern Engineering, University of Alaska Fairbanks, 1764 Tanana Loop, Fairbanks, AK 99775-5910, USA
7  British Antarctic Survey, High Cross, Madingley Road, Cambridge CB3 0ET, UK; maey@bas.ac.uk
*  Correspondence: steven.fassnacht@colostate.edu; Tel.: +1-970-491-5454

**Abstract:** We conducted an experiment comparing wind speeds and aerodynamic roughness length ($z_0$) values over three snow surface conditions, including a flat smooth surface, a wavy smooth surface, and a wavy surface with fresh snow added, using the wind simulation tunnel at the Shinjo Cryospheric Laboratory in Shinjo, Japan. The results indicate that the measurement location impacts the computed $z_0$ values up to a certain measurement height. When we created small (4 cm high) snow bedforms as waves with a 50 cm period, the computed $z_0$ values varied by up to 35% based on the horizontal sampling location over the wave (furrow versus trough). These computed $z_0$ values for the smooth snow waves were not significantly different than those for the smooth flat snow surface. Fresh snow was then blown over the snow waves. Here, for three of four horizontal sampling locations, the computed $z_0$ values were significantly different over the fresh snow-covered waves as compared to those over the smooth snow waves. Since meteorological stations are usually established over flat land surfaces, a smooth snow surface texture may seem to be an appropriate assumption when calculating $z_0$, but the snowpack surface can vary substantially in space and time. Therefore, the nature of the snow surface geometry should be considered variable when estimating a $z_0$ value, especially for modeling purposes.

**Keywords:** anemometric measurements; wind speed; snow bedforms; wind-blown snow; Shinjo Cryospheric Environment Laboratory

## 1. Introduction

A large portion of the earth is seasonally snow-covered for numerous months of each year [1,2]. The snowpack surface, as the interface with the atmosphere [3], is a spatially and temporally important boundary [4] that is quite dynamic [5–8] and greatly influenced by wind patterns [7,9,10]. The aerodynamic roughness length ($z_0$) [11–13] is an important metric for the snowpack surface, used for calculating latent and sensible heat fluxes [3,12,14]. Typically, the calculation of $z_0$ requires an anemometric profile to determine the variation in wind speed at differing heights above the surface [3,15]. There are a variety of methods for measuring the anemometric profile [16] and the resulting $z_0$ [13], which impacts the estimation of sublimation [17,18].

Snow surface $z_0$ values vary by several orders of magnitude, with the focus often being snow on glaciers [6,13,19] or on sea ice [12,14]. Brock et al. [6] summarized $z_0$ values using the wind profile method from the literature ranging from 0.2 to $14 \times 10^{-3}$ m. For a specific location, $z_0$ has been found to vary over time; on the Weddell Ice Sheet, $z_0$ varied from $10^{-6}$ to almost $10^{-1}$ m, partially dependent on the presence of blowing snow [5]. Due mostly to the snow depth, at a single location, the anemometric $z_0$ varied from 0.2 to $2 \times 10^{-3}$ m for a smooth underlying soil surface and then ranged from 1 to $40 \times 10^{-3}$ m when the soil was plowed [8].

Usually, meteorological stations constructed to measure the wind profile are established over homogeneous terrain, i.e., relatively flat surfaces and not among vegetation elements or obvious surface roughness features [3,20,21]. However, most natural surfaces, especially the snow surface, are heterogeneous in texture, which influences wind speed measurements and the computed $z_0$ value. Further, the snow surface shape can change rapidly as snow bedforms [22] can move at several meters per hour [23,24]. These snow bedforms have a period of 5 to 20 cm and can be small 0.2 to 2 cm high ripples or larger 5 to 18 cm high waves [23,25]. If we consider a fixed location meteorological tower and moving snow bedforms, then the snow surface changes at the tower, which is analogous to moving the tower to different snow surface shapes.

This paper examines the variation in calculated $z_0$ values obtained from fine vertical resolution (10 to 35 mm increments) anemometric measurements as a function of the sampling locations in a controlled environment (Table 1). We created a series of homogeneous bedforms in the shape of periodic, sinusoidal waves on a snow surface in a cold-room wind tunnel [26] and asked the following questions: (1) Are the wind profiles different based on horizontal sampling location? (2) At what height does the measurement profile converge regardless of the horizontal sampling location? (3) How does the computed $z_0$ value change as a function of the horizontal sampling location? And (4) how does the computed $z_0$ value change after fresh snow is blown onto the snow surface? Measuring the wind profile at different locations represents both a changing surface and spatial variability.

**Table 1.** A summary of the laboratory experiments (also see Figure 1d for wave position). For the snow wave experiment, $z_0$ was computed above the snow surface (2w, 2f, 2l, and 2t), as well as above the datum at the height of the trough (2wd, 2fd, 2ld, and 2td). For the furrow snow bedform (2f), $z_0$ was also computed without the bottom measurements (2fc and 2fdc).

| Experiment | Surface | Figure | Measurement Height (mm) |
|---|---|---|---|
| 1 | flat | Figure 1a | 35–385 by 35 mm increments |
| 2w | Bedform/wave–windward | Figure 1b | |
| 2f | Bedform/wave–furrow (top) | Figure 1b | |
| 2l | Bedform/wave–leeward | Figure 1b | |
| 2t | Bedform/wave–trough (bottom) | Figure 1b | |
| 2wd, 2fd, 2ld, 2td | Wind profile above common datum | Figure 1b | 10–210 by 10 mm, then 210–385 by 35 mm |
| 2fc, 2fdc | Furrow with bottom of profile clipped | Figure 1b | |
| 3w | Fresh snow-covered wave–windward | Figure 1e | |
| 3f | Fresh snow-covered wave–furrow (top) | Figure 1e | |
| 3l | Fresh snow-covered wave–leeward | Figure 1e | |
| 3t | Fresh snow-covered wave–trough (bottom) | Figure 1e | |

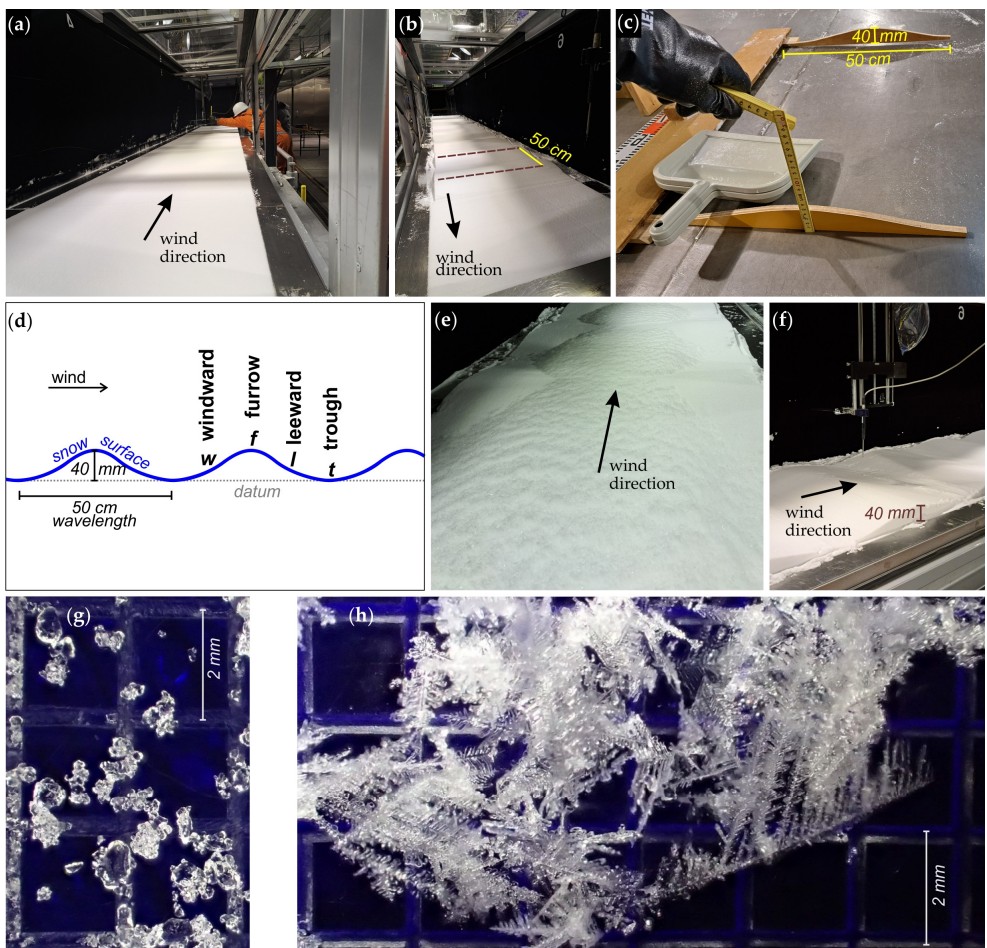

**Figure 1.** (**a**) Looking downwind in the CES wind tunnel over the flat snow surface; (**b**) looking upwind over the snow wave surface; (**c**) the form used to make the 50 cm long, 4 cm high snow bedforms; (**d**) conceptual diagram of the snow bedforms and the four relative horizontal sampling locations (w—windward, f—furrow, l—leeward, and t—trough); (**e**) looking downwind at the fresh snow-covered bedforms; (**f**) view of the snow bedforms and the hot-wire anemometer used to measure wind speed; (**g**) image of the old snow used to create the flat and snow bedform surfaces; and (**h**) image of the dendrite-shaped snow crystals (Type A) used as fresh snow blown across the snow surface.

## 2. Methods

This work was performed in the Cryospheric Environment Simulator (CES) of the National Research Institute for Earth Science and Disaster Resilience (NIED) in Shinjo, Yamagata Prefecture, Japan [26] (Table 1, Figure 1a,b,e,f). In the 14 m long, 1 m high, and 1 m wide CES wind tunnel, a flat snow surface was created using old snow that had a density in the range of 400–500 kg/m$^3$ (Figure 1a). The size of the wind tunnel allowed for a well-developed, mixed layer of air to flow across the snow surface. The cold room and wind tunnel were operated at a temperature of −10 °C, so the snow was also at a temperature of −10 °C.

Once the flat snow surface wind profile was measured, 10 snow bedforms (Figure 1b) were created on top of the flat snow surface using a 50 cm long, 4 cm high form (Figure 1c). The wind speed was then measured at four horizontal sample locations along one specific wave, 9 m longitudinally along the wind tunnel (experiments 2w, 2f, 2l, and 2t in Table 1 and Figure 1d). Measurements at different locations on the bedforms provide examples of both a spatio-temporally variable snow surface [7] and the movement of bedforms about a fixed anemometer tower [23–25].

Dendrite-shaped snow crystals (Type A) were created at CES (see Figure 1 by Abe and Kosugi [26] and Figure 4 by Nemoto et al. [27]). Using their technique, we also created these snow crystals in our experiment, which were then wind-blown across the surface of the bedforms (Figure 1e). The fresh snow density was measured to be 30 kg/m$^3$, as also observed by Abe and Kosugi [26]. Wind profiles were measured in the same four horizontal sampling locations over the fresh snow-covered bedforms (experiments 3w, 3f, 3l, and 3t in Table 1) and were used over the smooth bedforms in experiment 2 (Figure 1d, Table 1). This was not an investigation of snow bedforms and their movement e.g., [23,24], and as such, neither the shaped snow bedforms (Figure 1b,d,f) nor the fresh snow, once it was blown onto the bedforms, moved during the wind profile measurements.

Wind speed was measured using a Kanomax Climomaster Hot-wire Anemometer Model 6501 (https://kanomax-usa.com/products/climomaster-anemometer-6501-series/, last accessed 14 September 2023) (Figure 1f, Appendix A, and Table A1), which was affixed to an overhead hoist that could be adjusted horizontally and vertically. The anemometer was set to the lateral center of the wind tunnel and adjusted longitudinally to be at the desired horizontal sampling location along the wind tunnel. The initial vertical height was adjusted manually. Subsequent vertical heights were adjusted to within 0.1 mm from outside the wind tunnel.

For the flat snow surface, wind speeds were measured from 35 to 385 mm, every 35 mm vertically, above the flat snow surface, 9 m longitudinally along the wind tunnel (experiment 1 in Table 1). For the bedforms, the wind speed was measured every 10 mm vertically from the snow surface up to 210 mm and then every 35 mm up to 385 mm (experiments 2 and 3 in Table 1). Unless indicated otherwise, heights are measured above the snow surface (the height above the surface); these measurements are similar to measurements taken in the field after the snow surface has evolved, and the measurement if above the surface. Since the air is flowing across the surface, a height above a common datum was also selected; here, it was the height above the lowest vertical location on the snow bedform, specifically the trough. As such, the height above the datum increased the windward and leeward measurements by 20 mm and then the furrow measurement by 40 mm (Figures 1d and 2b).

To assess the variability in the anemometer measurements, the wind speed within the tunnel was run at 4 m/s until deemed stable (Appendix A), as per Abe and Kosugi [26]. Temperature (Figure A1) and wind speed (Figure A2) were recorded every second for 30 s, each at four different heights (10, 50, 100, and 200 mm above the surface). Both temperature and wind speed were deemed to be within the stated operating parameters (Table A1).

For each of the surfaces (Table 1), the wind speed was run at 4 m/s until the fluid (air) was deemed to be stable (Abe and Kosugi 2019) [26]. The wind profiles were plotted as a function of the measurement height, considering the datum to be the lowest point, or the trough (bottom) of the snow wave. The value of $z_0$ was computed using the wind speeds and the measurement height above the snow surface for each horizontal sampling location, assuming a natural logarithmic profile (Appendix C). The coefficient of determination ($R^2$) and the 95% confidence intervals for the slope and y-intercept were computed for each profile. The latter two were used to estimate the range of $z_0$ values with 95% confidence; these values were used to assess if $z_0$ values from different horizontal sampling locations were significantly different. When the profile deviated from a natural logarithmic shape, the outlier measurements were not included in the $z_0$ computation (for the measurements at the furrow, experiment 2f). The $z_0$ values were compared as a function of the horizontal sampling location (Figure 1d) and the absence or presence of blowing snow, as well as to the flat snow surface. The $z_0$ values were also computed using the height above the common datum, set at the trough (Figure 1d). The flat and bedform wind profiles were plotted above the measurement surface (Figure 2a) and above the common datum (Figure 2b).

For the bedforms and fresh snow blown onto the bedforms, there are wind speed measurements at 23 different vertical locations (Table 1 and Figure 2). These were systematically subset to recompute $z_0$ using fewer data points; every other point was removed, two of three points were removed, etc. The $z_0$ value computed using fewer points was

plotted versus the number of points used in the computation for windward measurements (experiments 2w and 3w in Table 1). The percentage difference was also computed.

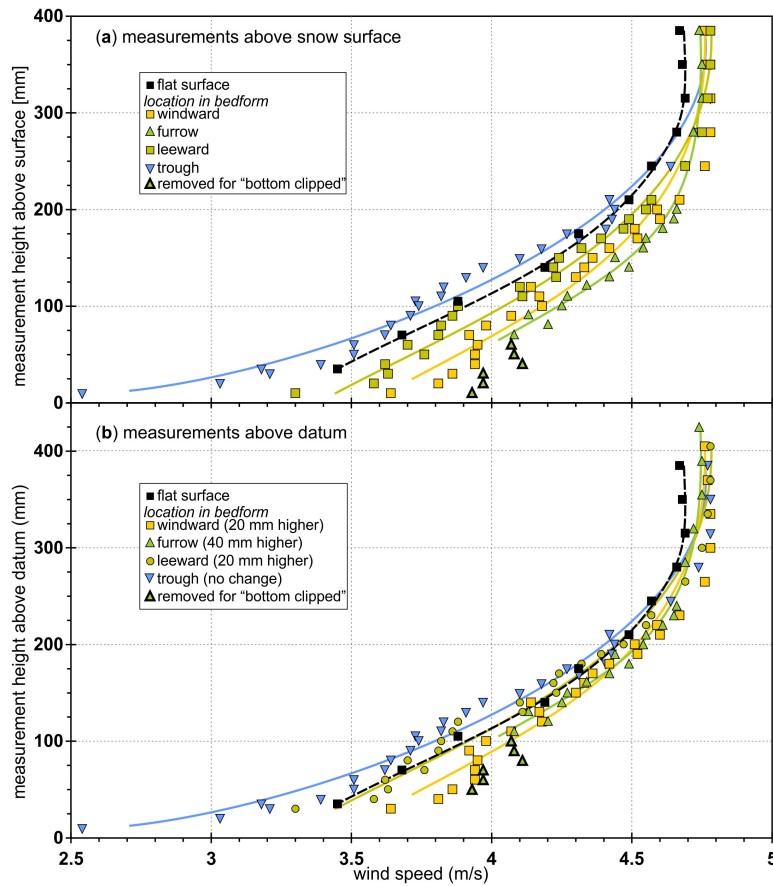

**Figure 2.** Wind profiles (measurement height versus wind speed) for the flat snow surface and the four horizontal sampling locations in the snow wave (windward, furrow or top, leeward, and trough or bottom) for (**a**) the height above the surface and (**b**) the height above the datum.

## 3. Results

When compared above the snow surface, the wind speeds are not similar along their profile, until more than 300 mm above the surface (Figure 2a). Above the datum, the wind speed increases with height above the surface more similarly, but not completely in a uniform natural logarithmic pattern (Figure 2b and $R^2$ values in Appendix B). The wind profiles have a different shape based on the horizontal sampling location (Figure 2b). The leeward sampling location was most similar to the flat surface, with the trough being the slowest and the furrow being the fastest at the lowest measurement heights (Figure 2b). Above the furrow, the wind speeds were faster at 70 and 80 mm above the datum (30 and 40 mm above the surface) than at 90 or 100 mm above the datum. The wind speeds among the four measurement locations were most different for the lowest 150 mm above the datum (Figure 3a), with the wind being fastest at the furrow, then at the windward location. The leeward and trough profiles were similar, with the slowest horizontal wind at the trough (Figure 3a).

For the windward location on the snow wave, wind speeds were essentially constant from 265 mm above the datum (245 mm above the surface) and higher (Figure 2b). For the trough, the wind speed was constant 320 mm above the datum (280 mm above the surface), while for the leeward and furrow locations, this height was 335 and 355 mm (315 mm above the surface), respectively (Figure 2b). At the furthest measurement heights above the surface, the wind was about 0.09 m/s slower on the flat surface compared to the snow bedforms (Figure 2b).

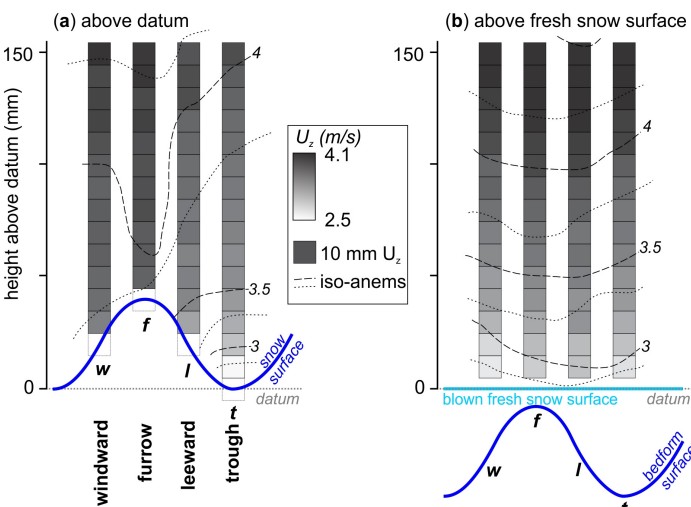

**Figure 3.** Wind speed from 10 to 150 mm above the snow surface at the windward, furrow, leeward, and trough measurement locations at 10 mm increments, with iso-anems (lines of equal wind speed) over the (**a**) bedform and (**b**) snow surface with fresh snow blown across it.

The addition of fresh snow onto the bedforms altered the wind profiles (Figure 3b), especially for the furrow and windward horizontal sampling locations (Figure 4). The lowest wind speeds were about 1 m/s slower for the same measurement height above the fresh snow-covered bedforms as compared to the smooth snow surface bedforms. At the leeward location, wind speeds were similar between the smooth bedforms and fresh snow-covered bedforms, with a maximum difference of about 0.5 m/s. For the trough, the wind speed measurements were almost the same for the two experiments, except at a few of the faster speeds, where values were almost 0.5 m/s faster for the fresh snow-covered wave surface (Figure 4).

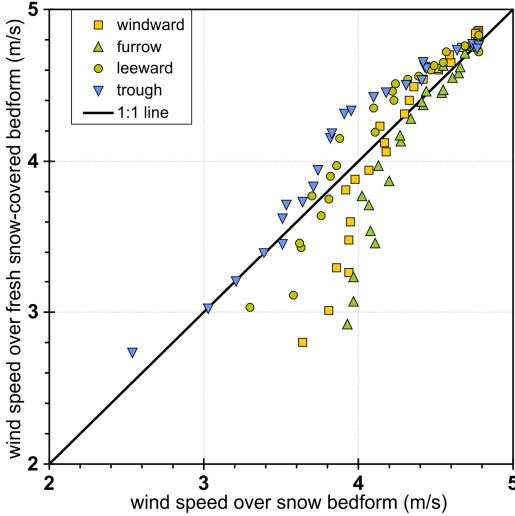

**Figure 4.** Comparison of the wind speed at all the vertical measurement locations over the four horizontal sampling locations on the snow bedforms. Here, wind speeds over the smooth snow surface bedforms are plotted versus wind speeds over the fresh snow-covered bedforms. Measurements were above the snow surface.

Direct calculations of $z_0$ on the snow bedform at a wind tunnel wind speed of 4 m/s varied from 4.27 to $5.75 \times 10^{-3}$ m (Figure 5 and Table 2) with 78 to 91% of the variance explained (Figure 2b and Table 2). Adjusting the vertical measurement height to a common datum decreased $z_0$ by 16 to 19% with the horizontal sampling location on the furrow

decreasing the most due to the largest change in height of 40 mm. For the three locations where the height was adjusted to a common datum, the explained variance increased by 5 to 8% (Table 2). The logarithmic fit was the poorest for the furrow horizontal sampling location wind profile (Figure 2b); removing the lowest six measurement heights (10 to 60 mm) decreased the $z_0$ by about 25% and increased the explained variance by 11% (to an $R^2$ of 0.91). We also considered an average datum at the mid-height of the bedforms, i.e., using the windward and leeward heights, which raises the datum for the trough by 20 mm and lowers the datum of the furrow by 20 mm (necessitating the removal of the two lowest measurements). This decreased the trough $z_0$ by 25% but did not change the furrow $z_0$.

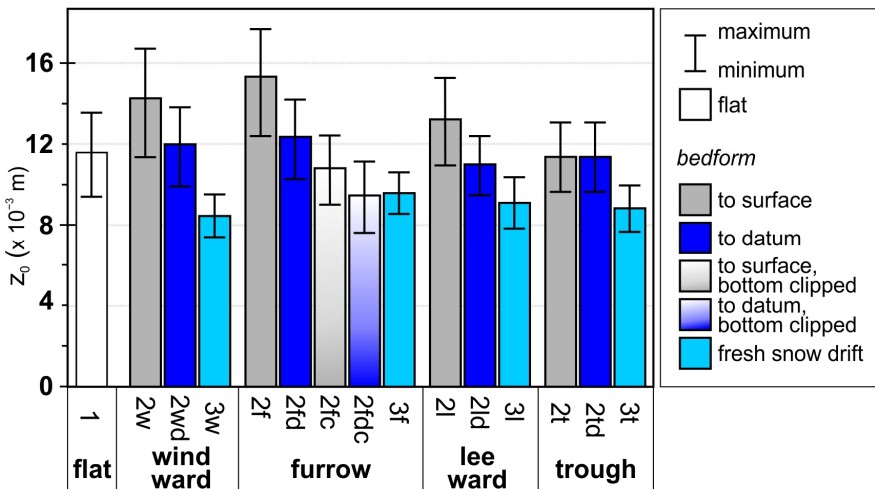

**Figure 5.** Distribution of $z_0$ values for the flat surface, and for the four horizontal sampling locations over both the smooth snow bedforms (experiment 2) and the fresh snow-covered bedforms (experiment 3), all with respect to the snow surface. Also shown are the $z_0$ computations using the height above the datum (2wd, 2fd, 2ld, 2td), as well as the values computed with the bottom of the furrow profile clipped or removed (2fc and 2fdc) (see Table 1 and Figure 2).

Fresh snow blown onto the snow bedforms decreased $z_0$ by between 22% (trough horizontal sampling location) and 41% (windward horizontal sampling location) (Figure 5 and Table 2). The logarithmic fit was very good ($R^2$ of 0.96 to 0.97). The variance in $z_0$ among the horizontal sampling locations was small, from 3.18 to $3.60 \times 10^{-3}$ m (Table 2).

The 95% confidence interval was computed for the $z_0$ values from the 4 m/s wind speed wind tunnel experiments (Figure 5 and Table 2). None of the computed $z_0$ values for any horizontal sampling location on the smooth snow bedforms or fresh snow-covered bedforms were significantly different than the flat surface $z_0$ (at the 95% confidence interval). When adjusting the $z_0$ computation to be above a common datum versus above the snow surface, the difference was not statistically significant. However, the $z_0$ values were significantly different between the horizontal sampling locations for fresh snow-covered bedforms versus the smooth snow bedforms, except not significant between the troughs (Figure 5 and Table 2). The $z_0$ values for the fresh snow-covered wave surfaces are generally not significantly different than the flat surface $z_0$ values.

Adding fresh snow onto the snow bedforms decreased $z_0$ by 22% (trough horizontal sampling location) and 41% (windward horizontal sampling location) (Figure 5 and Table 2). The logarithmic fit was very good ($R^2$ of 0.96 to 0.97). The variance in $z_0$ among the horizontal sampling locations was small, from 3.18 to $3.60 \times 10^{-3}$ m (Table 2).

As the amount of wind speed data used to compute $z_0$ decreases, the variation in $z_0$ compared to using all data increases (Figure 6). For the windward horizontal sampling location, the largest difference is using the fewest data points (three) (Figure 6a). It is possible to compute $z_0$ with only three (see 3w) or four (see 4w) wind speed data points that are almost the same as $z_0$ computed with the entire profile. It is best to have measurements at a distribution of heights. On average, using five wind speed measurements to estimate

$z_0$ is the same as using six or seven and yields similar results as using 8 to 11 wind speed measurements (Figure 6b).

**Table 2.** Computed anemometric-based aerodynamic roughness length ($z_0$) values for the different locations (also see Table 1 and Figure 1 for details on the locations) with the lower and upper limits at the 95% confidence interval and $R^2$ value. The $z_0$ values for fresh snow are denoted by *, and the clipped (see Figure 2) furrow $z_0$ values are denoted by #.

| Number | Form/Location | $z_0$ ($\times 10^{-3}$ m) | | | $R^2$ |
|---|---|---|---|---|---|
| | | **Mean** | **Low** | **High** | |
| 1 | Flat | 4.35 | 3.53 | 5.08 | 0.973 |
| *Above snow surface* | | | | | |
| 2w | Bedform: windward | 5.35 | 4.26 | 6.26 | 0.852 |
| 3f | Windward on fresh snow drift | 3.18 * | 2.78 | 3.57 | 0.972 |
| 2f | Bedform: furrow (all data) | 5.75 | 4.65 | 6.62 | 0.780 |
| 2fc | Furrow (bottom clipped) | 4.06 # | 3.38 | 4.66 | 0.908 |
| 3f | Furrow on fresh snow drift | 3.60 * | 3.21 | 3.98 | 0.973 |
| 2l | Bedform: leeward | 4.96 | 4.11 | 5.72 | 0.902 |
| 3l | Leeward on fresh snow drift | 3.42 * | 2.94 | 3.89 | 0.960 |
| 2t | Bedform: trough | 4.27 | 3.62 | 4.90 | 0.943 |
| 3t | Trough on fresh snow drift | 3.32 * | 2.88 | 3.74 | 0.970 |
| *Above datum set at trough height* | | | | | |
| 2w2 | Windward | 4.50 | 3.72 | 5.18 | 0.912 |
| 2f2 | Furrow (all data) | 4.64 | 3.85 | 5.32 | 0.866 |
| 2fdc | Furrow (bottom clipped) | 3.56 # | 2.86 | 4.18 | 0.899 |
| 2l2 | Leeward | 4.13 | 3.56 | 4.65 | 0.949 |
| 2t2 | Bedform trough | 4.27 | 3.62 | 4.90 | 0.943 |

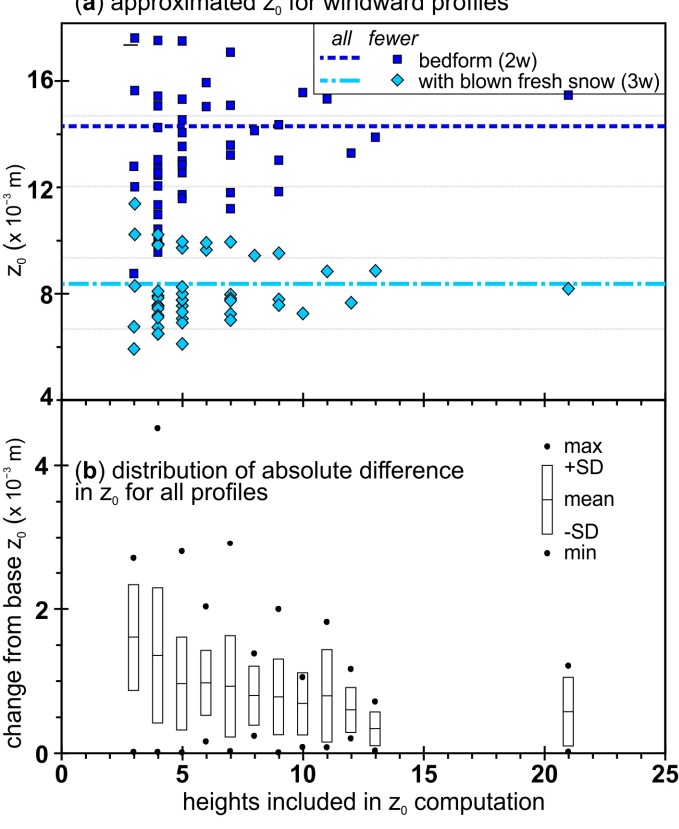

**Figure 6.** (**a**) Approximated $z_0$ values for the windward sampling on the bedform (dark blue) and for fresh snow blown onto the bedform (light blue) computed using a subset of the vertical wind speed data, and (**b**) the distribution of the absolute difference between $z_0$ computed using all wind speed data versus using fewer data for all bedform measurements (experiments 2 and 3).

## 4. Discussion

### 4.1. Experiments and Implications

We evaluated wind profiles (Figures 2 and 3) and computed $z_0$ (Figure 5 and Table 2) based on where they are measured (Figure 1 and Table 1). The wind profiles converged at about 300 mm above the datum, as also seen in the wind tunnel studied by Gromke et al. [28]. Wind profiles should be considered relative to a common datum (Figures 2b and 3) rather than above the surface (Figure 2a). This common datum could be an average of the roughness elements, especially when they are large like sastrugi [29]. Here, shifting the datum from the trough to the mid-point of the bedforms, i.e., raising it 20 mm, decreased the trough $z_0$ by 25% but did not change the other $z_0$ values. The measurement height of meteorological variables ($z$ in Equation (A1)) is the height of the sensor and must be with respect to the atmosphere-surface interface, i.e., the position of the snow surface. This must incorporate changes in snow depth due to accumulation, metamorphism–compaction, and ablation [18].

The values of $z_0$ for each location along the bedforms were in the same range as the flat snow surface (Figure 5 and Table 2). These are within the range of $z_0$ from the literature (Table 3). Snow surface roughness and, thus, $z_0$ varies spatially [6], temporally [7,19], and directionally [29]. Here, the computed differences in $z_0$ are at most half an order of magnitude, and not orders of magnitude (Table 3), and it sufficed to use three to five wind speed measurements to compute a reasonable $z_0$ (Figure 6); Jackson and Carroll [29] used five wind speed and direction measurements from 0.39 to 8.22 m, and eight air temperature sensors from 0.15 to 8.0 m above the snow surface. Due to the large variation in snowpack $z_0$ in the literature, we recommend collecting either vertical anemometric measurements [4,6,8,12,13,15,29] or surface geometry measurements [8,13,19] to attain an estimate of $z_0$ at any study site.

**Table 3.** Range of aerodynamic roughness length ($z_0$) values for snow from the literature, and in this study (Table 2 and Figure 5).

| $z_0$ Value (mm) | Conditions | Method | Citation |
|---|---|---|---|
| 0.001 to 100 | On ice sheet with some blowing snow | Wind profile | Andreas et al. [5] |
| 0.01 to 70 | Directional over sastrugi | Wind profile | Jackson and Carroll [29] |
| 0.17 to 0.33 | Fresh snow (mean of 0.24) | Geometry | Gromke et al. [28] |
| 0.2 to 2 | Snow on smooth surface | Wind profile | Sanow et al. [8] |
| 1 to 40 | Snow on plowed surface | Wind profile | Sanow et al. [8] |
| 3 to 25 | Snow on plowed surface | Geometry | Sanow et al. [8] |
| 5.5 | Snow on glacier ice | Geometry | Munro [13] |
| 4.4 | Flat snow surface | Wind profile | This study (1) |
| 4.1 to 4.6 | Locations along bedform | Wind profile | This study (exp. 2) |
| 3.2 to 3.6 | Fresh snow on bedforms | Wind profile | This study (exp. 3) |

The experiments were performed in a wind tunnel (Figure 1), where wind speed was controlled and stable [26]. In nature, while wind tends to be directional [30], any variation in wind direction over roughness elements on the snow [7] would influence $z_0$ [19,29]. Meteorological stations are usually established over flat, bare ground surfaces [20,21]. However, the seasonal snowpack surface that accumulates over the bare ground can vary substantially in geometry, both temporally and at various spatial scales. Therefore, the variable nature of the snow surface geometry [7,26] should be considered when estimating a $z_0$ value, even at locations that may have flat surfaces when there is no snow cover for part of the year, especially for modeling purposes [8,18,31].

We created homogeneous, sinusoidal snow bedforms, not to mimic reality but to create spatial variability. In nature, snow bedforms are not uniformly sized or spaced, and are asymmetrical [23,24]. It is possible to scale the snow bedforms from nature to the wind tunnel [32], but the proper shape would need to be modeled, as snow bedforms are steeper

on the leeward side [22,25]. Further, it is possible to have the smaller ripples form and move on top of the larger snow waves [22]. Since the mobile snow bedforms are self-organized [23,24], it is possible that, in nature, they are more aerodynamic and $z_0$ would be reduced. There are other less mobile wind-induced features like sastrugi [24,33,34] and ablation-induced features, like sun cups [35,36], penitents [35,37], or others [38]. These will create large turbulence across a snow surface and, thus, large $z_0$ values [6,34].

Here, we had fresh snow (Figure 1h) blown (Figure 1e) over the constructed snow bedforms (Figure 1b). However, during the measurement of the wind profile (Figure 1f), the fresh snow was not moving, thus not altering $z_0$ [39]. When snow is moving across the surface, this has been observed to increase the roughness [27,32,39]. In nature, snow grains fragment as they move across the surface [40]; thus, what is shown here is the initial movement of fresh snow (Figure 1e).

### 4.2. Measurements and Additional Data

The accuracy of the wind speed measurements is in the range of 0.1 m/s; the stated accuracy of the hot-wire anemometer is 0.015 m/s (Table A2), and the maximum observed variation over 30 s was 0.16 m/s (Figure A2). Wind speed was measured per 10 or 35 mm increase in height above the surface (Table 1), and it took five seconds or less to record wind speed. As such, the largest variation over any five-second period in the assessment of the hot-wire anemometer was 0.08 m/s. Further, at a height of 315, 350, or 385 mm above the surface of the snow bedforms, the measured wind speed varied by, at most, 0.04 m/s (Figure 2b). Therefore, the variations between the wind profiles for various horizontal sampling locations (Figure 2b) are based on actual differences in wind speed and not measurement discrepancies. Thus, more wind speed measurements are needed in varying terrains [16]. We also need to further assess the spatial interpolation of wind measurements [41–44].

The hot-wire anemometer (Figure 1f) measured horizontal wind speeds, parallel to the wind tunnel (Figure 1). Even in a controlled environment, such as a wind tunnel, wind flows are not all horizontal. In nature, measuring wind over the snow is difficult [16,45]. Future work should also assess the vertical wind component, which would assist in assessing roughness [4,46] and estimating components of the snow energy balance [16,47,48].

Since the nature of the snow surface is quite dynamic [6,7], and measurements of snow surface geometry are becoming more prevalent [19,49–51], more frequent temporal measurements of the snow surface [19,52,53] are recommended as the snowpack evolves during a winter season. The roughness of a snow surface varies over spatial and temporal scales [10], and an assortment of tools can be used to measure the different resolutions and extents [10,50,52,54,55].

Wind measurements are rarely taken at such a fine vertical resolution, as was performed in this study (10 to 35 mm, see Table 1), and, thus, may not capture all the variations in $z_0$. The snow bedforms created in the CES wind tunnel during our experiments in Shinjo, Japan (Figure 1b,f), were smaller than the features often seen in nature (e.g., sastrugi, sun sups, etc., as discussed above), so the resolution used herein is likely not necessary over larger snow features. Additionally, although eddy covariance is able to measure three-dimensional wind dynamics, it may not assess what is occurring between the sensor and the snow surface [16]. More measurements [56–59] should be considered at some research sites to ensure an appropriate representation of the snowpack surface variability [7,10,26] and other snowpack properties, such as snow depth [60,61]. Multi-sensor approaches using lidar and drones can also bridge scales [62–66], together with nested-scale in situ measurements to add a temporal component [67,68]. Satellite data will prove useful in the future, including across resolutions [66,69–71].

Here, the focus is only on the wind profile and, thus, $z_0$ for momentum (Table 2 and Figure 5). The sensible and latent heat flux equations use the temperature and humidity profiles, respectively. Specifically, each flux equation also uses a roughness length, i.e., $z_{0\text{-}T}$ for temperature (convection) and $z_{0\text{-}Q}$ for humidity (diffusion). These are usually set to

the value of $z_0$ for momentum, but this assumption is not true for aerodynamically rough surfaces [4,14]. Measurements of $z_{0-T}$ and $z_{0-Q}$ are limited [39] and additional data should be collected to solidify the correlation between the momentum, convection, and diffusion roughness lengths [4,12,14,39].

## 5. Conclusions

Wind profiles above a snow surface differ based on where they are measured, as seen in our wind tunnel experiments. Such measurements are relevant for understanding the snowpack and its energy balance and need to consider the dynamics of the snowpack using the height from the sensors to the average height of any roughness features. Computation of $z_0$ requires three and ideally at least five measurements of the wind profile. Due to the variation in snowpack $z_0$, it is more than a constant model parameter, and it should be estimated anemometrically or at least geometrically. Wind tunnel experiments and even outdoor scaling experiments (e.g., Tabler [32]) can be used to simulate the complex nature of the snowpack surface.

The computed aerodynamic roughness length in this study varied based on the horizontal sampling location for the smooth snow bedforms and, to a lesser extent, for the fresh snow-covered snow bedforms. The differences among the horizontal sampling locations were not statistically significant, but the addition of fresh snow did significantly alter the computed $z_0$. Overall, the largest difference in $z_0$ was 35% among the snow wave horizontal sampling locations (trough vs. furrow) and 13% for the fresh snow-covered snow bedforms (leeward vs. furrow). These variations in $z_0$ will impact the modeling of the snowpack energy balance, especially since snowpack $z_0$ appears to be dynamic. Therefore, the real estate expression "location, location, location" is relevant when measuring wind, and likely other meteorological variables; specifically, the geometric characteristics of the surface upwind and at the site must be considered. Further, any spatio-temporal variability in the convection and diffusion roughness lengths need to be better assessed.

**Author Contributions:** Conceptualization, S.R.F., K.S., M.N., J.E.S. and K.K.; methodology, M.N., S.R.F., K.S. and K.K.; formal analysis, S.R.F. and J.E.S.; investigation, S.R.F., K.S., M.N., J.E.S. and K.K.; data curation, S.R.F. and J.E.S.; writing—original draft preparation, S.R.F., K.S., M.N., J.E.S. and K.K.; writing—review and editing, S.R.F., K.S., J.E.S., M.M.F. and M.E.T.; visualization, S.R.F.; project administration, K.S. and S.R.F.; funding acquisition, K.S., S.R.F., M.N. and J.E.S. All authors have read and agreed to the published version of the manuscript.

**Funding:** The data were collected while S.R.F. was on a fellowship from the Japanese Society for the Promotion of Science <https://www.jsps.go.jp/> (last accessed 2 March 2023) (S-Fellowship S19145), hosted by the Japan Agency for Marine-Earth Science and Technology (JAMSTEC) <https://www.jamstec.go.jp/> (last accessed 2 March 2023). JAMSTEC provided administrative and related support during the culmination of this project. Funding for K.S. was provided by the Japan Society for the Promotion of Science (JSPS) KAKENHI grants (grant numbers 19H05668, 21H04934, and 22H03758) and the Arctic Challenge for Sustainability II (ArCS II) (program grant number: JPMXD1420318865). The data were collected at the Cryospheric Environment Laboratory, Snow and Ice Research Center, the National Research Institute for Earth Science and Disaster Resilience in Tokamachi, Shinjo-Shi, Yamagataken, Japan <https://www.bosai.go.jp/seppyo/index_e.html> (last accessed 14 September 2023). This research was indirectly funded by the U.S. Geological Survey National Institutes for Water Resources (U.S. Department of the Interior; grant number: 2019COSANOW; "The Dynamic Nature of Snow Surface Roughness" project), through the Colorado Water Center. M.M.F. was supported by core funding from the UK Natural Environment Research Council (NERC) to the British Antarctic Survey's Atmosphere, Ice and Climate Program.

**Institutional Review Board Statement:** Not applicable.

**Informed Consent Statement:** Not applicable.

**Data Availability Statement:** The data are included in Appendix B.

**Acknowledgments:** We thank K. Togashi, G. Okawa, and K. Suzuki at the Cryospheric Environment Laboratory for their assistance with the experiments. The fourth reviewer provided new insight into snowpack bedforms and a very thoughtful and thorough review; thank you.

**Conflicts of Interest:** The authors declare no conflicts of interest.

## Appendix A. Hot-Wire Anemometer Specification and Assessment

To measure the wind speed, we used a Kanomax Climomaster Hot-wire Anemometer Model 6501 (Table A1). The CES wind tunnel in Shinjo, Japan, was run until the fluid (air) was stable (Figure 1). To assess variability in the anemometer measurements, it was run for 30 s with measurements of temperature (Figure A1) and wind speed (Figure A2) recorded every second. This was performed at four heights: 10, 50, 100, and 200 cm above the snow surface (Figures A1 and A2).

**Table A1.** Stated specifications of the range, accuracy, and resolution of the Kanomax Climomaster Hot-wire Anemometer Model 6501 (https://kanomax-usa.com/products/climomaster-anemometer-6501-series/, last accessed 14 September 2023).

|  | **Air Velocity (m/s)** | **Temperature (°C)** |
|---|---|---|
| Range | 0.01 to 50.0 m/s | −20 to 70 °C |
| Accuracy | +/−2% of reading or 0.015 m/s (whichever is greater) | +/−0.5 °C |
| Resolution | 0.01 (from 0.01 to 9.99 m/s) | 0.1 °C |

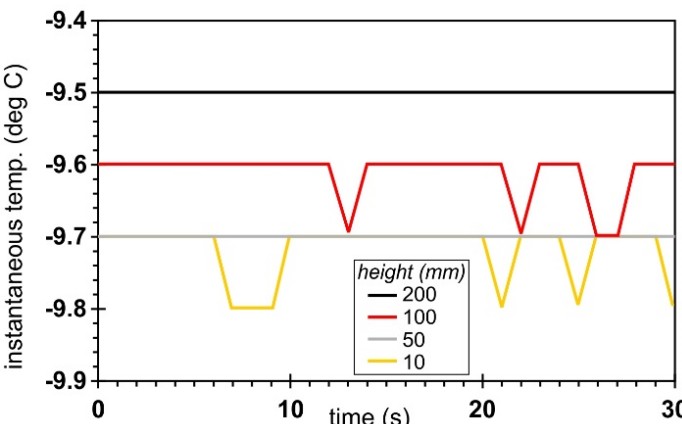

**Figure A1.** Time series of instantaneous temperature recorded every one second for 30 s testing interval.

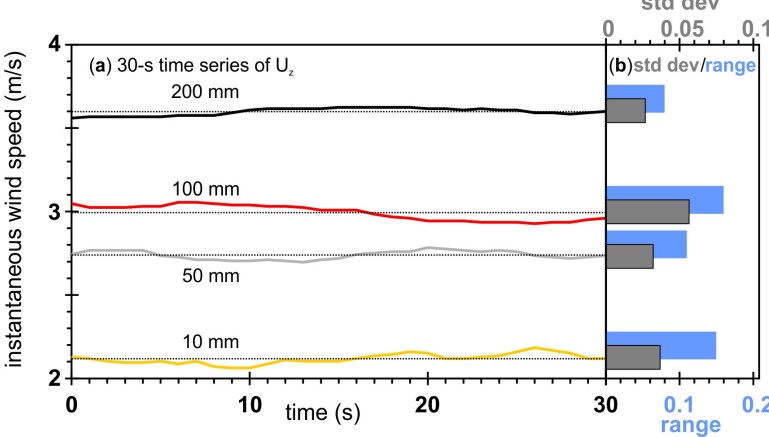

**Figure A2.** (**a**) Time series of instantaneous wind speed recorded every one second for 30 s testing interval, and (**b**) standard deviation (std dev) and range of wind speed at each of the four measurement heights. Dotted lines are the mean wind speed for each measurement height.

## Appendix B. Wind Speed Data Collected during the Experiments and Computed $z_0$ Values

This appendix presents the wind speed data collected at the different heights in the various horizontal sampling locations for the flat surface, snow wave, and fresh snow-covered snow wave, as outlined in Table 1 (Table A2).

**Table A2.** Wind speed in m/s as a function of height for flat snow surface, four locations in the wave, and four locations in the wave with blowing snow present.

| Height above the Surface (mm) | Flat | Wave Windward | Wave Furrow | Wave Leeward | Wave Trough | Blowing Snow in Wave Windward | Blowing Snow in Wave Furrow | Blowing Snow in Wave Leeward | Blowing Snow in Wave Trough |
|---|---|---|---|---|---|---|---|---|---|
| 10 | | 3.64 | 3.93 | 3.3 | 2.54 | 2.8 | 2.92 | 3.03 | 2.73 |
| 20 | | 3.81 | 3.97 | 3.58 | 3.03 | 3.01 | 3.07 | 3.11 | 3.02 |
| 30 | | 3.86 | 3.97 | 3.63 | 3.21 | 3.29 | 3.23 | 3.43 | 3.2 |
| 35 | 3.45 | | 4.18 | | 3.18 | | | | |
| 40 | | 3.94 | 4.11 | 3.62 | 3.39 | 3.26 | 3.46 | 3.46 | 3.39 |
| 50 | | 3.94 | 4.08 | 3.76 | 3.51 | 3.48 | 3.54 | 3.64 | 3.45 |
| 60 | | 3.95 | 4.07 | 3.7 | 3.51 | 3.6 | 3.71 | 3.77 | 3.62 |
| 70 | 3.68 | 3.92 | 4.025 | 3.81 | 3.535 | 3.81 | 3.77 | 3.75 | 3.71 |
| 80 | | 3.98 | 4.2 | 3.82 | 3.64 | 3.88 | 3.87 | 3.9 | 3.73 |
| 90 | | 4.07 | 4.13 | 3.86 | 3.71 | 3.94 | 3.97 | 3.97 | 3.83 |
| 100 | | 4.18 | 4.275 | 3.88 | 3.74 | 4.06 | 4.13 | 4.15 | 3.94 |
| 105 | 3.88 | | 4.11 | | 3.73 | | | | |
| 110 | | 4.17 | 4.27 | 4.11 | 3.82 | 4.12 | 4.17 | 4.19 | 4.15 |
| 120 | | 4.14 | 4.34 | 4.1 | 3.83 | 4.23 | 4.28 | 4.35 | 4.18 |
| 130 | | 4.3 | 4.42 | 4.23 | 3.91 | 4.31 | 4.37 | 4.4 | 4.31 |
| 140 | 4.19 | 4.33 | 4.410 | 4.22 | 3.955 | 4.4 | 4.39 | 4.46 | 4.33 |
| 150 | | 4.36 | 4.44 | 4.24 | 4.1 | 4.49 | 4.46 | 4.51 | 4.42 |
| 160 | | 4.42 | 4.54 | 4.32 | 4.18 | 4.54 | 4.45 | 4.54 | 4.45 |
| 170 | | 4.52 | 4.55 | 4.39 | 4.31 | 4.6 | 4.47 | 4.56 | 4.5 |
| 175 | 4.31 | | 4.44 | | 4.27 | | | | |
| 180 | | 4.51 | 4.61 | 4.47 | 4.41 | 4.61 | 4.55 | 4.6 | 4.53 |
| 190 | | 4.6 | 4.65 | 4.49 | 4.43 | 4.65 | 4.58 | 4.63 | 4.62 |
| 200 | | 4.59 | 4.66 | 4.55 | 4.44 | 4.7 | 4.62 | 4.65 | 4.61 |
| 210 | 4.49 | 4.67 | 4.55 | 4.57 | 4.42 | 4.75 | 4.63 | 4.72 | 4.65 |
| 245 | 4.57 | 4.76 | 4.69 | 4.69 | 4.64 | 4.79 | 4.71 | 4.76 | 4.73 |
| 280 | 4.66 | 4.78 | 4.72 | 4.75 | 4.74 | 4.85 | 4.76 | 4.75 | 4.77 |
| 315 | 4.69 | 4.78 | 4.75 | 4.77 | 4.78 | 4.86 | 4.78 | 4.74 | 4.76 |
| 350 | 4.68 | 4.77 | 4.75 | 4.78 | 4.78 | 4.85 | 4.76 | 4.72 | 4.76 |
| 385 | 4.67 | 4.76 | 4.74 | 4.78 | 4.77 | 4.84 | 4.76 | 4.83 | 4.74 |

## Appendix C. Computation of Anemometric-Based Aerodynamic Roughness Length

The value of $z_0$ measured by the vertical series of wind measurements is estimated from the logarithmic wind profile since the wind is assumed to be stable (Appendix A), using the following equation:

$$U_z = \frac{U^*}{\kappa} ln\left(\frac{z}{z_0}\right), \tag{A1}$$

where $U_z$ is the wind speed at measurement height $z$, $U^*$ is the shear velocity, and $\kappa$ is the von Kármán constant. At one specific point in time and space, $U^*$ is constant, so Equation (A1) can be inverted to solve for $U^*$. Here, we have 11 to 30 values of $U_z$ at unique $z$ (Table A2). We compute the slope ($m$) and y-intercept ($b$) between pairs of $U_z$ and $ln(z)$ measurement points, then compute $exp(m/b)$, where $exp()$ is the exponential function to yield $z_0$ (Table 2). We also compute the correlation coefficient ($R^2$) between the $U_z$ and $ln(z)$ measurement points (Table 2).

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
