# Peer review of "Location Dictates Snow Aerodynamic Roughness"

_2813-8740_

Round 1
Reviewer 1 Report
Comments and Suggestions for Authors
The proposed manuscript reports the results of an interesting laboratory study on the effect of the measurements of aerodynamics roughness lenght on the wind profile with different snow surface conditions. The main result is that the nature of the snow surface geometry should be considered when estimating the aerodynamic roughness lenght.
The paper is clear and the results are of interest.
Comments on the Quality of English Language
Language must be checked becuase several typos can be founad in the manuscript.
Author Response
Comments and Suggestions for Authors:The proposed manuscript reports the results of an interesting laboratory study on the effect of the measurements of aerodynamics roughness lenght on the wind profile with different snow surface conditions. The main result is that the nature of the snow surface geometry should be considered when estimating the aerodynamic roughness length.
The paper is clear and the results are of interest.
> no revision necessary.
Comments on the Quality of English Language:Language must be checked because several typos can be found in the manuscript.
> we have gone through the paper and reviewed the language.
Reviewer 2 Report
Comments and Suggestions for Authors
This study used wind tunnel simulations to study the aerodynamic roughness of snow. The author compared the wind speed and z0 values under three different snow conditions, including a flat smooth surface, a wavy smooth surface, and a wavy surface with fresh snow added. Results indicate that the measurement location impacts the computed z0 values up to a certain measurement height. However, the authors could do much better job on analyzing and presenting the data before publishing. Followings suggestions still need to be improved:
1. The author should provide experimental parameters such as snow density and test temperature.
2. The corresponding theoretical calculation formula is missing in the text. The author needs to provide a calculation method and formula for aerodynamic roughness length.
3. Is there a difference between Z0 in Figure 4 and z0 in the manuscript? Generally, z0 is used to represent the aerodynamic roughness of a smooth surface, and Z0 is the aerodynamic roughness of a covered surface.
4. In the results section, the author focuses more on the description of multiple experimental data and lacks in-depth theoretical analysis.
5. It is recommended that the author refer to the following literature and conduct a detailed analysis of their research content.
Kikuchi T. A wind tunnel study of the aerodynamic roughness associated with drifting snow. Cold Regions Science and Technology, 1981, 5(2):107-118.
Gromke C B, Walter B, Manes C, et al. Aerodynamic roughness lengths of snow. Boundary-Layer Meteorology, 2011, 141(1):21-34. DOI:10.1007/s10546-011-9623-3.
Author Response
- The author should provide experimental parameters such as snow density and test temperature.
> The density of the old snow was in the range of 400-500 kg/m3, and new snow is in the range of 30-50 kg/m3 (Abe and Kosugi, 2019). Here only the fresh snow density was measured, and it was 30 kg/m3.
> Does the reviewer mean snowpack temperature? The temperature of the wind tunnel is given (Appendix A), as the cold room was at a constant temperature of -10C (+/- 0.25C), and since the first snow layer was thin (~ 3 cm thick), it can be assumed to be at -10C.
> This information was added to the text.
- The corresponding theoretical calculation formula is missing in the text. The author needs to provide a calculation method and formula for aerodynamic roughness length.
> We believe that this is understood by the community. However, we have added this into an Appendix.
- Is there a difference between Z0 in Figure 4 and z0 in the manuscript? Generally, z0 is used to represent the aerodynamic roughness of a smooth surface, and Z0 is the aerodynamic roughness of a covered surface.
> We don’t understand the difference in the reviewer’s symbols (Z0 vs. z0)? Is there a difference between a smooth surface (e.g., snow?), and a covered surface (covered in snow)? Figure 4 has a lower case z0.
- In the results section, the author focuses more on the description of multiple experimental data and lacks in-depth theoretical analysis.
> I think that the review means the Discussion section. The Results have been revised. The Discussion has been rewritten to focus more on the implications of this work, rather than just an interpretation of the work.
- It is recommended that the author refer to the following literature and conduct a detailed analysis of their research content.
Kikuchi, T. A wind tunnel study of the aerodynamic roughness associated with drifting snow. Cold Regions Science and Technology 1981, 5(2), 107-118. https://doi.org/10.1016/0165-232X(81)90045-8
Gromke, C. B.; Walter, B.; Manes, C.; Lehning, M.; Guala, M. Aerodynamic roughness lengths of snow. Boundary-Layer Meteorology 2011, 141(1), 21-34. https://doi.org/10.1007/s10546-011-9623-3
> These papers are relevant, and we now cite them and additional papers.
Reviewer 3 Report
Comments and Suggestions for Authors
This is a very interesting study, and its results are useful for calculating latent and sensible flux. There are a few shortcomings and problems of this manuscript, as follows:
In the introduction section, literature review is not comprehensive. It is suggested that the authors should review the relevant studies on the measurement of snow aerodynamic roughness.
In the discussion section, it is suggested that the authors should present the results of other researchers on snow aerodynamic roughness in a table, so that readers can more clearly compare the similarities and differences between the authors’ and other researchers’ results.
In addition, the citation format in the text does not conform to MDPI's journal format.
Author Response
In the introduction section, literature review is not comprehensive. It is suggested that the authors should review the relevant studies on the measurement of snow aerodynamic roughness.
> We have included more specifics on the range of aerodynamic roughness length values observed in the literature. We have not added exhaustive literature review, as that would make the paper unnecessarily long. However, we have expanded the literature cited.
In the discussion section, it is suggested that the authors should present the results of other researchers on snow aerodynamic roughness in a table, so that readers can more clearly compare the similarities and differences between the authors’ and other researchers’ results.
> Brock et al. (2006) provides a decent summary of z0 values in the literature. We have added more citations to present the range of z0, and a table.
In addition, the citation format in the text does not conform to MDPI's journal format.
> the MDPI format is difficult to use when writing a paper (and in my opinion not user friendly). We will alter the citation format later.
Reviewer 4 Report
Comments and Suggestions for Authors
Attached as pdf. Good paper overall!
My comments are intended to improve the paper’s readability and broaden its potential readership without placing undue burden on the authors.

Author Response
We appreciate all the effort that this reviewer put into reading and reviewing our paper. It provided a new perspective and we incorporated almost all of their comments.
Good paper overall!
> We appreciate all the effort that this review put into assessing this paper. We have considered all of their comments and incorporated them as best as possible.
Summary
The authors present novel measurements of snow aerodynamic roughness that will be relevant to snow scientists, atmospheric boundary scientists, and field scientists installing weather stations in snowy locations. The paper is clearly written and describes a well-designed experiment.
My comments are intended to improve the paper’s readability and broaden its potential readership without adding undue burden on the authors.
General comments
I believe the key implication of this paper for users of wind speed data is that measurements of wind speed and aerodynamic roughness over snow surfaces are expected to vary in space and time, on 0.1-20m length scales and minutes-to hours time scales, if the wind speed and snowfall are great enough to make bedforms, and if the anemometer is lower than 3.5+ m.
> agreed.
The variation in time is not yet drawn out in this paper, but is a clear implication: snow waves move, so a fixed anemometer will sometimes be on the windward side of the nearest wave, and sometimes on the leeward side. This variation might be periodic in a wind-tunnel, but is probably intermittent in nature where wave speeds vary greatly with time, snow supply, and wind
The length and time scales are set by the sizes of bedforms in nature, which include:
- Waves, of wavelength 3-20m, heights of 5-18cm, and speeds up to 6+ m/h
- Ripples, of wavelength 5-20cm, heights of 0.2-2cm, and speeds up to 3+ m/h (and likely much higher, as the limited literature shows small ripples traveling faster than big waves)
(Measurements combined from
https://agupubs.onlinelibrary.wiley.com/doi/full/10.1002/2015JF003529 and
https://tc.copernicus.org/articles/13/1267/2019/ )
The anemometer height is drawn from this paper’s Fig. 2, where measured wind speeds over different parts of the wave seem to converge above ~3.5m (the authors should check this against their data!). Above this height I presume the bedforms no longer have measurable effects on the measured wind speed. I would guess that the required height for convergence would be higher in particularly windy/snowy places, where strong winds and large snow volumes can make bedforms taller than 4cm.
I think this takeaway will be meaningful for anyone working with wind speed measurements taken over snow surfaces and, would like to see it included in the abstract and discussion. Please note that this is not prescriptive, and I encourage the authors to consider their data and amend my statements if needed to fit the facts.
Filhol, S.; Sturm, M. Snow bedforms: A review, new data, and a formation model. J. Geophys. Res. Earth Surf. 2015, 120, 1645-1669. https://doi.org/10.1002/2015JF003529
> the bedforms are 4 cm high and did not move. Fresh snow was blow onto the snow waves, and they did not move during the wind profile measurements. The convergence of the wind profiles was at 0.35 m. We have added text in the Introduction and Discussion to add some focus on snow bedforms. This is not the focus of this work, but we appreciate that the review has made these detailed suggestions.
Detailed comments
Abstract
L19 – I would re-structure the abstract to lead with, “We conducted an experiment comparing wind speeds and z0 values over three snow surface conditions…” (this is the unique part of the paper)
> agreed.
L31 – I would add “vary substantially in space and time” for completeness
> changed.
L35 – Consider replacing keywords “snow waves” with “snow bedforms” and “fresh snow” with “wind-blown snow” depending on target audience
> changed. Through most of the text, the term wave was replaced with bedform.
Introduction
L39 – First two sentences are very general; consider removing them and starting with “A large portion of the earth is seasonally snow-covered...”
> this has been changed.
L46 – Consider rephrasing for clarity: “The aerodynamic roughness length (z0) is an important metric for the snowpack surface, used for calculating latent and sensible [heat] fluxes.”
> changed.
L48-67 – Very clear!
> Thank you.
L49 – Missing bracket on citation
> added.
L68 – I would add information here on what snow-waves look like in nature, including size ranges and the fact that they move. See comments and references in the ‘general comments’ section.
> we have added some text to put snow bedforms into context and explain their use in this study.
Methods
L94 – FYI, I’ve included some comments on the methodology and its similarity to real snow forms in the discussion
> we have altered the text to explain that the measurements at various points in the snow bedforms represents both spatial/temporal variability and the movement of bedforms about a fixed tower.
L103 / Figure 1
- Could you add the height and wavelength measurements to Fig. 1d?
> This has been added.
- Could you also show the measurement positions and ‘datum height’ clearly in Fig. 1d? See comment on L130.
> This has been added.
- It would be useful to see a photograph with the side profile of the wave against a dark background (if you have one – no need to re-set experiment for this)
> Unfortunately, we do not have one for some reason. Next time, we will ensure that we get more such photographs. We have added the dimensions to several of the photographs.
L130 – Could you add a brief description of your height measurements? It took me a while to understand the heights vs. datum heights. This is necessary for interpreting Figs 2, 3, and 4 and Table 2 and needs to be easy to understand. I would suggest:
- Consider using phrases like “height above snow surface” and “height above ground/fixed level” throughout the paper – this would be simple to understand and would help anyone who sees the figures in isolation
> We have tried to use this protocol throughout, in order to be consistent.
- Add a short description to the methods, like: “Unless indicated otherwise, heights are measured above the snow surface; these measurements will be similar to field measurements taken by a field scientist who arrives after the wave has formed and measures from its surface. “Datum heights / heights above ground” are measured above a fixed point, in this paper the trough of the wave; these measurements will be similar to those taken by a fixed anemometer anchored in the ground below the snow.”
> Thank you. A version of this has been added to the methods.
- Consider showing the datum lines / measurement points graphically, e.g. as an addition to Fig. 1d.
> This has been added.
- See also comments on L161
> x
Results
L131 / Figure 2 – super clear!
> thank you
L131 / Results – I would add a comment on the height at which snow waves cease to impact measured wind speed, or (better), a side panel on Fig 2 that shows “variance of measured wind speed with position relative to wave” against height.
> We have added an additional figure of the wind profiles versus the height above the snow surface.
L135 / Results – As a general comment, I found it surprising that wind speeds were sometimes higher with waves / roughness than without – in nature I would expect winds over a rough surface to be slower / lose more momentum. I assume this is a function of the wind-tunnel setup?
> Others found that the distribution of the wind is influenced by the wind-tunnel characteristics. Here we limited our measurements to about 40% of the height of the wind-tunnel. Similar to Gromke et al. (2011), our wind profiles converged around 300 mm above the datum.
L135 / Results – My biggest unanswered question here is “how does the spatial average z0 value change when there are waves vs. on flat snow?”. Could you have a go at answering this? Maybe displaying the estimate graphically in Fig 4?
> The values of z0 for each location along the bedforms were in the same range as the flat snow surface. This is presented in Figure 4 (now Figure 5).
L151-157 and Figure 3 – I think the takeaway from Fig. 3 is that drifting snow alters wind speed/roughness most at the furrow and least at the trough. However, I’m not sure if wind speed is the most valuable comparison here, given that the wind tunnel is (presumably?) set to keep wind speed constant – I think it would be more valuable to express this in terms of z0 values and momentum loss from the air to the snow.
> xThese are presented in Figure 5 and Table 2. A new figure (3) has been added to show the relative wind distribution. Figure 3 (now 4) shows the variations in the profile, as per Figure 2.
Personally I might replace Fig. 3 with a simple bar chart showing “z0 with drifting snow / without drifting snow”, or remove it to supplemental material.
> These are in Figure 5.
L160 – Explained [by…]
> The new Figure 3 gives some insight into the differences among the wind profiles.
L161 – Bunch of questions about the choice of reference height. I think it’s worth writing a few lines about this to give future researchers a guide to select the best height for their measurements.
- How did you decide to set the common datum to the trough?
> The lowest surface height was set as the common datum, as the air is moving across the surface so we also want to evaluate the wind at the same horizontal locations. Any datum could have been chosen, but the using the lowest one made the most sense.
- Do the values of z0 change if your reference height is the average snow height instead?
> This brings the trough up 20 mm and the furrow down by 20 mm (and removes the two lowest measurements). This changes the furrow z0 by a minute among, but as with the other datum adjustments, decreases the trough z0, in this instance by 25%. This was added to the results section.
- Does the quality of the logarithmic fit and the need to clip bottom measurements change if you choose a different reference height (esp. for the furrow, which had both the biggest adjustment and worst fit)?
> Changing the datum actually increased the logarithmic fit. This was stated in the text as a 5 to 8% improvement in explained variance.
- Which measurement(s) do you think might reasonably used by people in the field?
> We should be measuring above the surface, so that we have a true value of z, as per Hultstrand and Fassnacht (2018), but we consider meteorological towers fixed, so we typically consider the height above the ground, which is incorrect, and yields different results. We analyzed the impact of decreasing the number of wind speed measurements to compute z0 (now Figure 6), and conclude that five measurements should be sufficient, but in some cases (depending on which locations), three yields a similar results.
L171 / Figure 3 -
> x
L198 / Table 2 and L190 / Figure 4 – I would group the data points here by height measurement (datum adjusted / not), rather than by position along wave. As noted in my comments on L130 I think the use cases are different and variation within each measurement group is likely more useful than variation between wave positions. Leaving it up to the authors’ preference.
> We agree that the table may be easier to read by grouping the above surface value and the above datum values, so Table 2 has been reorganized. Since we used color in Figure 4, we have left it as is, so that the reader can compare different locations (same color) and different datums, etc. (together).
Discussion
L198 / Discussion – Somewhere in here add the material from ‘general comments’ above
> We have rewritten much of the Discussion to add some
L198 / Discussion – Somewhere in the discussion add a paragraph on the differences between the experiment and typical field conditions, including:
- Wave size – 50cm wavelength / 4cm height is an uncommon size in nature, being much bigger than a typical ripple but smaller than a typical wavelength
> We have added a discussion on these differences, and that we can scale the bedforms from the field to the lab, as per Tabler (1980).
- Wave shape – Experimental waves appear to be perfectly sinusoidal. Real snow waves tend to be asymmetric, steeper on the leeward side, with an angular crest, and possible ripples on the windward side (https://tc.copernicus.org/articles/13/1267/2019/ and https://www.jstor.org/stable/1775469 ). I would bet that this shape difference has a significant impact on z0, as the waves are self-organized; my guess is that more aerodynamic shapes with lower z0 would be more stable and common in nature.
> This is likely more important/relevant than size, as size can be somewhat scales. We have added this to the Discussion.
- Snow crystals – Snow waves form during high-wind events; tiny ice spheres are probably more common than dendrites blowing over them, except in the first movements of fresh snow on older existing waves (https://agupubs.onlinelibrary.wiley.com/doi/full/10.1002/2017GL073039 )
> We have added this point to the discussion.
L198-270 / Discussion – In general, I think the paper will be more usable/citeable if you get a bit more specific about the implications of your work (though of course, as per the discussion above, it’s only an approximation of reality. In particular:
- How much uncertainty are bedforms likely to add to measurements of z0?
> We measured about 40% variation, but in reality this is likely much larger, i.e., orders of magnitude. We recommend tracking how the surface changes.
â—¦ How much variation do they add in your paper?
> About 40%, so something, but not an order of magnitude.
â—¦ How big is the range of measurements of z0 on snow in literature (e.g. from the 1994 paper cited on L265 – that’s a cool data point)?
> We added a table (3) with some of the data from the literature. Eddy Covariance is “gospel” for field measurements of fluxes and thus enabling the estimation of z0. However, …
â—¦ Can you make any reasonable guesses about variation in nature based on sizes of natural bedforms?
> Perhaps. We likely need to scale the bedforms (Tabler 1980), but that is beyond the scope of this paper.
- How many measurements are required to capture the “true” value of z0? How would you distribute them in space and time? How does the answer vary with measurement height?
> We assess this (Figure 6) and conclude five.
L258-262 – Optional: I think you could do a quick extra analysis on vertical resolution (either here or in the results section): drop some of your measurements to simulate “coarser” resolution, then do a plot of vertical resolution against apparent z0. See if anything interesting comes out?
> Thank you. See Figure 6.
Conclusions
271 / Conclusions – Somewhere in here add takeaway from ‘general comments’ above
> We have rewritten the Conclusions with takeways.
L272-275 – Very clear!
> thank you
L276-277 – Be specific; which differences would not be captured by coarser resolution, and importance of vertical vs horizontal coarseness
> This has been deleted, as the Conclusions have been rewritten, and are now more specific.
L278/279 – Finer resolution is not necessary for what? (I think this page belongs in the last paragraph of the discussion)
> This has been deleted, as the Conclusions have been rewritten, and are now more specific.
L280-287 – very clear!
> thank you
Appendix
Thanks for including complete data and measurement accuracy!
L339 / Table B-1 – I don’t think it affects your results, but why are the measurements at different heights for different experiments?
> initially we did the measurements at 35 mm intervals, then decided to change to 10 mm intervals up to 210 mm above the surface.
Good luck with the publication!
> Thank you!
Round 2
Reviewer 2 Report
Comments and Suggestions for Authors
The author has completed the relevant modifications and has no further suggestions for publication.
Author Response
thank you for your review
Reviewer 3 Report
Comments and Suggestions for Authors
The citation format in the text does not conform to MDPI's journal format. Other issues I pointed out have effectively been resolved.
Author Response
thank you for your review.